# Effect of Arbuscular Mycorrhizal Fungi and Rock Phosphate on Growth, Physiology, and Biochemistry of Carob under Water Stress and after Rehydration in Vermicompost-Amended Soil

**DOI:** 10.3390/metabo14040202

**Published:** 2024-04-03

**Authors:** Abderrahim Boutasknit, Mohamed Ait-El-Mokhtar, Boujemaa Fassih, Raja Ben-Laouane, Said Wahbi, Abdelilah Meddich

**Affiliations:** 1Multidisciplinary Faculty of Nador, Mohammed Premier University, BP 300, Nador 62700, Morocco; abderrahim.boutasknit@gmail.com; 2Center of Agrobiotechnology and Bioengineering, Research Unit Labelled CNRST (Centre AgroBiotech-URL-7 CNRST-05), Abiotic and Biotic Constraints Team, Cadi Ayyad University (UCA), Marrakesh 40000, Morocco; bo.fassih@gmail.com (B.F.); wahbi@ucam.ac.ma (S.W.); 3Laboratory of Agro-Food, Biotechnologies and Valorization of Plant Bioresources (AGROBIOVAL), Department of Biology, Faculty of Science Semlalia, Plant Physiology and Biotechnology Team, Cadi Ayyad University (UCA), Marrakesh 40000, Morocco; 4Laboratory of Biochemistry, Environment & Agri-Food URAC 36, Department of Biology, Faculty of Science and Techniques—Mohammedia, Hassan II University of Casablanca, Mohammedia 20000, Morocco; 5Laboratory of Environment and Health, Department of Biology, Faculty of Science and Techniques, Moulay Ismail University, BP 509, Errachidia 52000, Morocco; benlaouaneraja@gmail.com

**Keywords:** biostimulants, forest tree, symbiotic microorganisms, (in)-organic amendments, rehydration, tolerance

## Abstract

In the Mediterranean region, reforestation programs record failures following successive drought periods. The use of different plant-growth-promoting amendments and the understanding of drought-induced physiological and biochemical responses of carob will contribute to the reforestation program’s success. In this study, the effects of arbuscular-mycorrhizal-fungi (AMF), vermicompost (VC), and rock phosphate (RP) on carob seedlings under drought stress (DS) and recovery (REC) conditions were evaluated. A greenhouse experiment was conducted with carob seedlings grown in the presence of AMF, VC, and RP, applied alone or in combination under well-watered (WW), DS (by stopping irrigation for 12 days), and recovery (REC) conditions. The obtained results indicated that the triple combination (AMF + VC + RP) presented the highest improvement in water potential, photosynthetic pigment content, stomatal conductance, and chlorophyll fluorescence compared to the controls under DS and after REC. In addition, this combination resulted in improved tolerance of carob seedlings to DS and a high potential for rapid recovery after rehydration due to a high accumulation of sugars, proteins, and antioxidant enzymes. In summary, the results underline the importance of inoculating carob with AMF in combination with (in)-organic amendments in improving its tolerance to DS and its recovery performances.

## 1. Introduction

Long periods of successive drought, caused by climate change, are now becoming one of the major and most severe abiotic stress factors affecting plant growth and causing a decline in crop productivity in arid and semi-arid regions [1]. Soil moisture limitation affects hydraulic and nutritional conductivity, mainly impacting phosphorus (P), potassium (K), nitrogen (N), ion homeostasis, and gas exchange; it damages photosynthetic pigments, and reduces the photochemical efficiency of photosystem II (PSII) [2,3,4]. In addition, drought stress (DS) alters the antioxidant defense system and cell membrane stability by generating reactive oxygen species (ROS) accumulation and thereby oxidative stress [5]. Plants exposed to drought show various altered processes including disrupted water relations between tissues, gas exchange mechanisms, and symbiosis behavior, and reduced plant growth and adaptive capacity [6,7,8]. To maintain a plant’s water balance, stomata act as regulators to control the flow of gases between the leaf and the atmosphere [9,10]. The reduction in stomatal aperture leads to a reduction in the diffusion of carbon dioxide (CO_2_) at the leaf level, consequently leading to a decline in photosynthetic performances [11]. During the prolonged period of DS, changes in leaf water status and chlorophyll pigment concentration, as well as photosynthetic activity lead to variations in the osmotic adjustments through the accumulation of primary metabolites such as soluble sugars and proteins [3]. Drought also threatens forest sustainability in arid and semi-arid zones by limiting forest trees’ survival and reforestation success. The use of pioneer trees remains the most recommended solution for successful reforestation and coping with conditions of prolonged DS [12].

The carob tree is considered an essential element of the arboreal flora and forest ecosystem of the Mediterranean basin due to its tolerance to abiotic stress including drought and its adaptation to soil nutrient depletion [13,14,15]. The carob tree was widely cultivated due to its many nutritional benefits such as its richness in dietary fibers, minerals, polyphenols, tannins, and thermotolerant volatile organic compounds [16,17,18]. Carob leaves and pods have numerous pharmaceutical and cosmetic properties. Recently, carob extracts have been shown to exhibit antioxidant, antifungal, antibacterial, anti-inflammatory, antidiarrheal, and antidiabetic activities, as well as antiproliferative effects [19,20]. Furthermore, carob could be used in the prevention of free radical-related diseases and in neurodegenerative disorder therapy as a natural dietary supplement [21]. Additionally, it is used in the biotechnological fields of the food industry as a natural bioactive and in the cosmetic industry where carob seeds are used for their antioxidant/preservative function [22].

Beneficial microorganisms such as arbuscular mycorrhizal fungi (AMF) and (in)organic amendments including vermicompost (VC) and rock phosphate (RP) are considered among the potential strategies and solutions to boost plant tolerance under environmental stress in the reforestation programs [4,23,24,25,26]. AMF could improve forest vegetation establishment based on their ability to cope with harsh environmental conditions [27]. Several studies indicate that mycorrhizal symbiotic relationships improve plant performance through increasing root hydraulic conductivity, osmotic adjustment, stomatal conductance, and the host plant’s antioxidant systems, particularly under DS [28,29,30,31]. Furthermore, the application of VC as an organic amendment, rich in stable humic substances and mineral elements, has been shown to improve the soil’s structural, physico-chemical, hydric, nutritional, and biological properties [32,33,34]. In addition, it can enhance plant adaptation and tolerance mechanisms to drought, and significantly improve microfauna and microorganism growth and development, as well as soil enzyme activities [35,36]. Combining AMF with VC is an effective strategy to mitigate the drought alteration of cell morphology, physiology, and metabolism and to ensure stable, safe, and sustainable agricultural production [4,37].

Inorganic soil improvers are an important source of inorganic RP, which can be used as a natural fertilizer to promote plant growth and productivity, and stimulate enzymes and phosphate solubilization activities by soil microorganisms. On the other hand, it has been reported that AMF have the ability to secrete a number of organic acids that help to solubilize inorganic P. In addition, AMF possess specific enzymes called phosphatases and phytases that hydrolyze inorganic phosphates into the orthophosphate form assimilated by plant roots [38,39,40]. Recent studies have also demonstrated the ability of AMF to solubilize unavailable forms of P into available forms for plant uptake [41]. To our knowledge, no study has explored the impact of combining AMF with (in)-organic amendments (VC and RP) on carob growth and tolerance performances under prolonged DS and after recovery conditions. The aim of the present study was to evaluate the effects of different combinations of AMF, VC, and RP on morphological, physiological, nutritional, and biochemical traits of carob seedlings subjected to DS and after the rehydration phase (recovery). Furthermore, this study will uncover technological methods for successful reforestation, the improvement of forest trees’ survival, and boosting the productivity of carob trees under DS conditions.

## 2. Materials and Methods

### 2.1. Biological Materials, Application of Fertilizers, and Experimental Design

Carob seeds of the southern (coastal) ecotype from the Essaouira region (31.5085° N, 9.7595° W), Morocco, were obtained from the Institut National de la Recherche Agronomique (Marrakesh, Morocco). They were scarified with concentrated sulfuric acid for 30 min, rinsed, submerged in sterile distilled water for 24 h, and then germinated on moist filter paper in Petri dishes incubated at 28 °C for five days. The germinated carob seeds were transferred to trays containing sterilized peat. After two months, the carob seedlings were transplanted into plastic pots (27 cm length and 12 cm width) containing 3 kg of sterilized soil (at 0.11 MPa and 121 °C for 2 h). The used soil contains 1% organic matter (OM), 8 mg Kg^−1^ available P (AP), 900 mg Kg^−1^ N, 2356 mg Kg^−1^ calcium (Ca), and 568 mg Kg^−1^ available K with a pH value of 8.2, and an electrical conductivity (EC) of 0.14 mS cm^−1^. The soil texture is characterized by 51% sand, 19% clay, and 30% silt.

During transplanting, AMF-treated carob seedlings received 60 g of an AMF inoculum composed of spores, mycorrhizal roots, and soil in the proximity of their root system. The AMF inoculum was propagated from a consortium containing a mixture of 26 species of Glomales (*Rhizophagus clarus*, *R. diaphanum*, *R. intraradices*, *Funneliformis mosseae*, *F. geosporum*, *Acaulospora denticulata*, *A. kentinensis*, *A. spinosa*, *Claroideoglomus etunicatum*, *Glomus proliferum*, *Septoglomus constrictum*, *Diversispora epigeae*, *Glomus sp1*, *Glomus sp2*, *Glomus sp3*, *Glomus sp4*, *Glomus sp5*, *Acaulospora sp1*, *Acaulospora sp2*, *Acaulospora sp3*, *Acaulospora sp4*, *Gigaspora sp1*, *Gigaspora sp2*, *Gigaspora sp3*, *Entrophospora sp1*, and *Scutellospora sp1*). Non-inoculated carob seedlings received an equal amount of a filtrate obtained by passing 60 g of mycorrhizal inoculum dissolved in 20 mL of distilled water through 15–20 µm filter paper (Whatman, GE Healthcare, Buckinghamshire, UK).

The horse manure-based VC used in this study had the physicochemical properties shown in Table 1. When the seedlings were transplanted, 5% (*w*/*w*) of VC was added to the VC-amended pots.

The used RP was obtained from Office Chérifien des Phosphates (OCP), a Moroccan state-owned phosphate rock mining company. The RP had the following characteristics: AP: 20.60 mg kg^−1^; EC: 0.56 mS cm^−1^; pH 8.45; total organic carbon: 0.64%; and total OM: 1.10%. When the seedlings were transplanted, 1% (*w*/*w*) of RP was added to the RP-treated pots.

The eight treatments in the experimental design are as follows: (i) control, (ii) AMF, (iii) vermicompost (VC), (iv) rock phosphate (RP), (v) combination of AMF and VC (AMF + VC), (vi) combination of AMF and RP (AMF + RP), (vii) combination of VC and RP (VC + RP), and (viii) combination of AMF, VC, and RP (AMF + VC + RP). Treated carob seedlings were maintained at 75% field capacity (FC) and arranged in a completely randomized block design. Each treatment had 20 replicates for a total of 160 seedlings. After six months of regular irrigation at 75% FC and before the application of DS, growth, physiological, water status, and biochemical parameters were assessed on one third of the seedlings from each treatment (the first harvest). Then, the remaining two-thirds of the carob seedlings were subjected to DS, which consisted of a total cessation of watering for 12 days. After 12 days of DS, we measured the same parameters on half of the seedlings under DS during the second harvest. Then, the remaining carob seedlings under DS were regularly rehydrated for 5 days. After 5 days of recovery (REC), the third harvest was carried out measuring the same parameters as before. Carob seedlings were grown in a semi-controlled greenhouse with an average temperature of 26 °C, an average natural luminosity of 410 µmol m^−2^ s^−1^, and an average relative humidity of 68–70%. The position of the carob seedlings in the greenhouse was changed each month to minimize the potential effects of local or variable microclimatic conditions during the experimentation period.

### 2.2. Carob Growth and Root Mycorrhizal Colonization Measurement

Plant shoot height (SH), root length (RL), and shoot (SDW) and root (RDW) dry weights (dried at 75 °C until weight remained constant) were recorded after each harvest. Roots were washed with water and cut into 1-cm fragments. These fragments were placed in 10% KOH for 2 h and incubated at 90 °C. They were then acidified with 5% hydrochloridric acid (HCl)for 20 min. The fragments were then stained with 0.05% (*w*/*v*) trypan blue for 30 min at 90 °C [42]. Root mycorrhizal colonization was observed using a Zeiss Axioskop 40 microscope at 400× magnification. The mycorrhizal frequency (F) and intensity (I) were assessed on randomly selected 1 cm-long root fragments (20 fragments per glass slide) with five replicates for each treatment. F and I parameters were calculated according to McGonigle et al. [43] as follows:$$F\left(\%\right)=100\times \left(\frac{infected\;root\;segments}{root\;segments\;total}\right)$$
$$I\left(\%\right)=((95\times n5)+(70\times n4)+(30\times n3)+(5\times n2)+1\times n)/total\;root\;segments$$
where n is the number of fragments assigned with the index 0, 1, 2, 3, 4, or 5, with the following infection rates: 100 > n5 > 90; 90 > n4 > 50; 50 > n3 > 10; 10 > n2 > 1; 1 > n1 > 0.

### 2.3. Phosphorus Content Measurement

To assess leaf P content, samples were dried in an oven at 80 °C for 48 h, then finely ground using a grinder. The ground leaves were then incinerated in an oven at 550 °C for 6 h. The obtained ash was collected and digested in 3 mL 6N HCl, then evaporated on a hot sand plate. The residual fraction was reconstituted in hot distilled water. P mineral content was measured using the colorimetric method based on Olsen and Sommers [44] method.

### 2.4. Measuring Physiological Performance

Leaf water potential (Ψ_Leaf_) of carob seedlings was measured using a pressure chamber (SKPD 1400, Skye Instruments, Powys, UK) before dawn (06:00–08:00 a.m.). Mature and fully developed leaves were selected from the same range of the upper stem. Ψ_Leaf_ of each of the five plants per treatment was measured on the same days and immediately after the gas exchange measurements.

Stomatal conductance (gs) was measured on the abaxial surface of mature and well-developed carob leaves, from the same row of the upper part of the plant, between 9:00 and 11:00 a.m. on sunny days, using a portable porometer (Leaf Porometer LP1989, Decagon Device, Inc., Washington, DC, USA). Five measurements were taken per treatment.

Chlorophyll fluorescence (Fv/Fm) measurements were carried out on the youngest and most-developed leaves of the upper third row of the seedling using a fluorometer (Opti-sciences OSI 30p, Hudson, NY, USA). The selected leaves were acclimatized in the dark for 30 min using clips before measuring Fv/Fm on a 12.5 mm^2^ surface. Measured Fv/Fm corresponds to the quantum yield (Fv/Fm = (Fm − F0)/Fm), where Fm and F0 are the maximum and initial quantum yields of dark-acclimated leaves, respectively.

Photosynthetic pigments including chlorophyll a (Chl a), chlorophyll b (Chl b), total chlorophyll (Chl T), and carotenoids were extracted from fresh leaf samples (from the same row of the upper part of the plant) in 80% acetone as described by Arnon [45]. Leaf samples were previously crushed in liquid N immediately after harvest. An amount of 50 mg of crushed leaves was mixed with 4 mL of 80% acetone. The mixture was centrifuged at 10,000× *g* for 10 min. The optical density (OD) of the supernatants was measured at 480, 645, and 663 nm using a UV-vis spectrophotometer (spectrophotometer UV-3100PC). The following formulas were used to calculate photosynthetic pigment concentrations:$$Chlorophyll\;a\;\left(\frac{mg}{g}\right)=\left[\left(12.7\times OD663\right)-\left(2.69\times OD645\right)\right]\times \frac{V}{1000}\times DW$$
$$Chlorophyll\;b\;\left(\frac{mg}{g}\right)=\left[\left(22.9\times OD645\right)-\left(4.68\times OD663\right)\right]\times \frac{V}{1000}\times DW$$
$$Total\;Chlorophyll\;\left(\frac{mg}{g}\right)=[OD480+\left(0.114\times OD663\right)-(0.638\times OD645)]\times \frac{V}{1000}\times DW$$
where V = final volume of the extract, and DW = dry weight.

### 2.5. Measurement of Leaves Biochemical Traits

Fully expanded leaves of carob seedlings grown under WW, DS, and REC conditions were harvested at the end of the light period, flash-frozen, and ground to fine powder in liquid N. The determination of total soluble sugars (TSS) was carried out on powdered frozen leaves (0.1 g) mixed with 4 mL ethanol (80% *v*:*v*) [46]. The obtained supernatant (0.2 mL) was added to 0.25 mL phenol and mixed with 1.25 mL sulfuric acid. The OD of the mixture was measured after 15 min at 485 nm.

Total soluble protein content and antioxidant enzyme activities were assessed by homogenizing 0.25 g of frozen leaf powder with 1 M phosphate buffer (pH 7) containing 5% polyvinylpolypyrrolidone. The resulting mixture was then centrifuged at 18,000× *g* for 15 min at 4 °C. The obtained supernatant was used to determine protein content and antioxidant enzyme activity [47]. Total soluble protein was determined by the Bradford method [48], using bovine serum albumin (BSA) as a standard.

The activity of superoxide dismutase (SOD) was evaluated according to the method described by Beyer and Fridovich [49]. A unit of SOD activity has been defined as the ability to inhibit 50% of the photochemical reduction of *p*-nitroblue-tetrazolium (NBT) at 25 °C. SOD activity was expressed in unit min^−1^ mg protein^−1^. Peroxidase (POX) activity was measured using the method described by Tejera García et al. [47]. The reaction mixture consisted of K_2_HPO_4_/KH_2_PO_4_ buffer (100 mM), guaiacol (40 mM), H_2_O_2_ (10 mM), and enzyme extract (0.1 mL). Measured activity was expressed in unit mg protein^−1^. Measurement of polyphenoloxidase (PPO) activity was assessed by monitoring catechol oxidation at 410 nm, as described by Gauillard et al. [50]. The used solution consisted of K_2_HPO_4_/KH_2_PO_4_ buffer (100 mM, pH 6), catechol (50 mM), and enzyme extract (0.1 mL). PPO activity was expressed in mg protein^−1^ units.

Malondialdehyde (MDA) concentration was determined by the method of Savicka and Škute [51]. Lipid peroxides were extracted from 0.25 g of frozen powder subsamples and mixed with 10 mL of 0.1% (*w*/*v*) trichloroacetic acid (TCA). After centrifuging the extract at 18,000× *g* for 20 min, 1 mL supernatant mixture was added to 2.5 mL thiobarbituric acid (TBA), resulting in chromogen formation. The supernatant was then incubated at 95 °C for 30 min and the reaction stopped by placing the tubes in an ice bath. The OD was measured at 450, 532, and 600 nm. MDA content was calculated as follows:[*MDA*] = 6.45 × (*OD* 532 − *OD* 600) − 0.56 × *OD* 450.

The amount of hydrogen peroxide (H_2_O_2_) was measured using the method described by Velikova et al. [52]. Frozen crushed leaves (0.25 g) were homogenized in 5 mL TCA 10% (*w*/*v*). The mixture was then centrifuged at 15,000× *g* for 15 min. The supernatant (0.5 mL) was mixed with 0.5 mL of potassium phosphate buffer (10 mM, pH 7) with the addition of 1 mL potassium iodate (1 M). After 1 h incubation in the dark, the OD was read at 390 nm and H_2_O_2_ concentrations were determined using a standard H_2_O_2_ curve.

### 2.6. Statistical Analysis

Data for each phase were analyzed using SPSS Statistics software (version 23, IBM Analytics, Amsterdam, The Netherlands). A multivariate analysis of variance (MANOVA) was performed to test the effects of AMF, RP, and VC and their interaction on the measured parameters. The honest Tukey test was used at a significance level of 5% (*p* ≤ 0.05). Measurements were averaged over six independent replicates for growth and physiological parameters and three replicates for biochemical parameters. The mean values and standard errors were calculated.

## 3. Results

### 3.1. Mycorrhizal Colonization and Plant Growth Performance

Regardless of the applied conditions, a clearly distinct trend emerged between the different applied treatments (Table 2). F was higher in carob seedlings inoculated with AMF alone and/or in combination with RP (AMF + RP) and RP + VC (AMF + VC + RP) under the stress condition (Table 3). Similarly, carob seedlings inoculated with AMF and amended with RP alone and/or in combination with VC showed a higher I for the different phases. Indeed, the highest I was recorded in AMF + RP-treated seedlings, irrespective of the applied water regime. After the recovery phase, the F and I of carob seedlings showed an improvement of 9 and 8%, respectively, under the AMF + RP treatment, and of 5 and 2%, respectively, under the AMF + VC + RP treatment in comparison with the mycorrhizal seedlings under the same water regime. The interactions among AMF × DS and AMF × REC were significant for these two parameters (*p* < 0.001) (Table 2). In addition, the interactions among RP × DS, RP × REC, AMF × RP × DS, and AMF× RP × REC were significant for F (*p* < 0.05), while the interaction among AMF× VC × REC was significant for I (*p* < 0.001) (Table 2).

Under the WW condition, inoculated and/or amended carob plants showed a clear improvement in above-ground (SH and SDW) and below-ground (RL and RDW) growth traits compared with the control plants (Table 3). Significant effects of the combined treatments on growth traits were observed irrespective of the irrigation conditions of the carob plants. After 12 days of DS and five days of REC, the growth parameters measured showed no significant difference between carob seedlings and their counterparts under WW conditions. In fact, after the application of 12 days of DS, the increment in SDW and RDW were greater in plants treated with AMF + VC + RP, by 128 and 170%, respectively, than in the control under the same condition. The SDW and RDW accumulation of carob seedlings was enhanced by the application of the triple combination under DS conditions, compared to the control plants. Furthermore, the addition of organic (VC) and/or inorganic (PR) amendments to AMF-inoculated seedlings showed a significant attenuation (*p* < 0.05) of the deleterious effects of DS on growth and biomass accumulation under the REC condition (Table 2 and Table 3). The application of the triple combination under prolonged conditions of DS and after REC resulted in significant differences (*p* < 0.05) in biomass accumulation compared with seedlings amended or inoculated separately. The interaction among AMF× VC × REC was significant for carob seedlings SDW and RDW (*p* < 0.001) (Table 2).

### 3.2. Phosphorus Uptake Efficiency of Carob Seedlings under Drought and Recovery Regimes

To assess the nutritional status of carob plants under the different treatments, we analyzed leaf P levels under different regimes of irrigation (Table 3). P concentrations in carob leaves significantly varied according to the applied water regime (*p* < 0.05) and the applied treatment (*p* < 0.05) (Table 2). Regardless of the applied water regime, carob seedlings significantly increased P levels through AMF, RP, and VC treatments and their combinations compared with untreated seedlings. Under DS, application of AMF + RP, AMF + VC, and AMF + VC + RP induced greater accumulation of P levels in the leaves of carob seedlings by 52, 52, and 57%, respectively, compared with the untreated seedlings under the same condition. Furthermore, we found that the application of AMF + RP, AMF + VC, and AMF + VC + RP after rehydration resulted in an increase of 70, 55, and 65%, respectively, compared to the control seedlings under the REC condition. It should be noted that no obvious difference in P content was recorded after five days of recovery compared to the corresponding treatments under the DS condition.

### 3.3. Water Potential and Physiological Efficiency of Carob Seedlings under Drought and Recovery Regimes

Ψ_Leaf_ was around −0.5 MPa, under WW conditions, and no significant differences were noticed in inoculated and/or amended carob plants compared with the control plants (Figure 1a). Under the DS condition, Ψ_Leaf_ was significantly reduced in treated and untreated carob seedlings (*p* < 0.001) (Table 2), but this reduction was more pronounced in the absence of the applied fertilizers. In contrast, Ψ_Leaf_ was increased, in the presence of DS, by around 20% in seedlings treated with AMF + VC + RP and by 17% in those treated with VC + RP compared with the control seedlings subjected to prolonged DS for 12 days (Figure 1a). After five days of REC, we observed that treated seedlings showed a much greater ability to return to their initial state than untreated ones. Application of the VC + RP and AMF + VC + RP resulted in 61 and 69% recovery compared with the control seedlings under the WW condition. The interactions among RP × DS and VC × RP × DS were significant for Ψ_Leaf_ (*p* < 0.001). However, the interaction among RP × REC was significant for this parameter (*p* < 0.01) (Table 2).

Physiological traits related to photosynthesis showed a significant difference (*p* < 0.001) (Table 2) in gs (Figure 1b) and Fv/Fm (Figure 1c) between inoculated and/or amended and untreated carob seedlings, irrespective of the applied irrigation conditions. However, the application of the 12-day watering stop induced a severe decline in gs and Fv/Fm in treated and untreated carob seedlings. Under the DS condition, the extent of gs decline was greater in the control carob seedlings (52%), followed by the AMF + RP-treated plants (33%), and then AMF + VC + RP-treated plants (26%) compared to the corresponding treatments under the WW condition. Under DS, seedlings treated with AMF + RP and AMF + VC + RP increased gs by around 62 and 86%, respectively, compared to the control seedlings under the same condition. At recovery, the gs of the tripartite combination treated plants showed a greater ability (75%) to return to the initial state when compared with the same treatment under the WW condition. Similarly, after 12 days of water restriction, Fv/Fm was reduced to a greater extent in the untreated carob seedlings by 28% compared with their counterparts under WW conditions. Compared with the control seedlings under DS, the application of AMF + RP, AMF + VC, and AMF + VC + RP resulted in an increase in Fv/Fm by 28, 25, and 30%, respectively, under the same water regime. After the recovery phase, seedlings inoculated with AMF and amended with RP and VC showed a greater ability (88%) to bring Fv/Fm back to the control values when compared with the same treatment under the WW condition. Consequently, carobs treated with AMF and RP and/or VC showed higher values when under stress conditions and a greater ability to recover after a prolonged period of drought than carobs inoculated or amended separately. The interactions among AMF × DS, AMF × REC, VC × DS, and VC × REC (*p* < 0.001), AMF × VC × REC (*p* < 0.01), and AMF × RP × REC (*p* < 0.05) were significant for gs (*p* < 0.001). However, the interactions among AMF × DS and AMF × REC (*p* < 0.001), VC × DS (*p* < 0.01), and VC × REC (*p* < 0.05) were significant for Fv/Fm (*p* < 0.01) (Table 2).

### 3.4. Photosynthetic Pigment Content in Carob Seedlings under Drought and Recovery Regimes

The application of DS significantly affected photosynthetic pigment content in treated and untreated carob seedlings (Figure 2 and Table 2). Chl a (Figure 2a), Chl b (Figure 2b), Chl T (Figure 2c), and carotenoid (Figure 2d) contents were reduced in the control plants by 30, 49, 46, and 36% under DS conditions, compared with their corresponding levels under the WW condition. However, this reduction was less pronounced in seedlings treated with AMF + RP, AMF + VC, and AMF + VC + RP for Chl b (39, 48, and 47%, respectively), for Chl T (44, 42, and 41%, respectively), and for carotenoids (21, 22, and 18%, respectively), compared to the corresponding treatments under the WW condition. After five days of rehydration, chlorophyll pigment levels were significantly increased and at a higher level in carob seedlings treated with AMF + RP, AMF + VC, and AMF + VC + RP compared to the control seedlings under the same condition. These increases were 69, 59, and 80% for Chl a, 99, 95, and 114% for Chl b, 82, 79, and 89% for Chl T, and 24, 27, and 37% for carotenoids in AMF + RP, AMF + VC, and AMF + VC + RP, respectively. Photosynthetic pigment content was more affected in untreated stressed carob seedlings than in treated stressed ones. Furthermore, our data showed that the combination of AMF with RP and/or with VC had a better response under DS and REC conditions than treatments based on AMF, RP, and VC applied alone. The interactions among AMF × DS and AMF × REC were significant for all photosynthetic pigments (*p* < 0.001). However, the interactions among AMF × VC × DS, AMF × VC × REC, and AMF × RP × REC (*p* < 0.001) were significant for Chl b (*p* < 0.01). In addition, the interactions among AMF × RP × DS and AMF × VC × REC (*p* < 0.05), RP × DS and AMF × VC × RP × REC (*p* < 0.01), and AMF × VC × RP × DS were significant for Chl T (*p* < 0.001). Furthermore, the interaction among AMF × VC × RP × DS was significant for carotenoids (*p* < 0.001) (Table 2).

### 3.5. Protein and Total Soluble Sugar Contents in Carob Seedlings under Drought and Recovery Regimes

The effect of the application of the different water regimes and fertilizers on the protein and total soluble sugar (TSS) contents of carob seedlings was highly significant (Table 4). A determination of carbohydrate content in the leaves during DS and REC conditions revealed that plants inoculated with AMF and amended with PR and/or VC significantly (*p* < 0.001) increased TSS content compared with the corresponding treatment under the WW condition. After 12 days of DS, this TSS accumulation was greater in inoculated and amended carob seedlings, reaching 10% for AMF + RP, 6% for AMF + VC, and 14% for AMF + VC + RP in comparison to the untreated plants under the same condition. Similarly, under the DS condition, protein content was increased in carob leaves by 34% in AMF + RP, 30% in AMF + VC, and 41% in AMF + VC + RP compared to the control under the same water regime. After rehydration, the most significant increases in protein content were recorded in AMF + RP (30%), AMF + VC (33%), and AMF + VC + RP (31%) compared to the control seedlings under the REC condition. The interactions among AMF × DS, AMF× REC, and VC × RP × DS were significant for proteins (*p* < 0.05) (Table 2).

### 3.6. Hydrogen Peroxide and Malondialdehyde Contents in Carob Seedlings under Drought and Recovery Regimes

Under the DS condition, the oxidative stress marker (H_2_O_2_ and MDA) content sharply increased regardless of the applied treatment (Table 4). This increase was significantly more pronounced in the control carob seedlings than in untreated ones, irrespective of the applied water regime. Compared to the WW condition, DS-stressed seedlings showed higher H_2_O_2_ levels, particularly in the untreated seedlings by 38% and those treated with RP (37%), and AMF (23%). Similarly, MDA levels recorded a significantly greater increase in VC-treated seedlings (29%) followed by AMF + RP (25%), and then RP (23%) compared to the control seedlings under the DS condition. However, this accumulation was lower in carob seedlings treated with AMF, RP + VC, AMF + VC, and AMF + VC + RP. Furthermore, the elevated H_2_O_2_ and MDA levels following the exposure of carob seedlings to DS for 12 days were largely reduced after the REC phase. H_2_O_2_ and MDA concentrations decreased to lower levels in inoculated and/or amended seedlings than in untreated seedlings. After REC, seedlings treated with AMF + RP recorded the greatest decline in MDA content (11%) while VC recorded the greatest decline in H_2_O_2_ content (26%) compared with the corresponding treatments under the DS condition.

### 3.7. Antioxidant Enzymes Activity in Carob Seedlings under Drought and Recovery Regimes

The cessation of irrigation for 12 days induced a substantial increase in SOD, POX, and PPO activities in seedling leaves compared with WW conditions and regardless of the applied treatment (Figure 3). During the DS phase, AMF + VC + RP induced the greatest increase in SOD activity (9%), POX activity (32%), and PPO activity (23%). Nevertheless, after the REC phase, a decrease was recorded in SOD, POX, and PPO activity compared with activities recorded under DS conditions. During the REC phase, the SOD activity of inoculated and/or amended carob seedlings attained levels closer to their corresponding levels recorded under WW conditions, while POX and PPO activity were still relatively higher.

## 4. Discussion

Reforestation in arid and semi-arid Mediterranean regions remains limited, if not non-existent, due to unfavorable conditions associated with delayed rainfall and soil degradation [53,54,55]. In addition, soils in these regions are characterized by low water retention capacity, lack of organic matter, nutrient deficiency, and reduced microbiological biodiversity [56,57]. For successful reforestation in these unfavorable regions, soil quality needs to be enhanced by the addition of organic and inorganic amendments and beneficial microbes such as AMF. On the other hand, the use of plants adapted to unfavorable pedoclimatic conditions remains one of the most recommended solutions. The carob tree can be considered a water-limitation-tolerant species due to its numerous adaptation mechanisms such as the optimization of water use strategies [14,55,58], thus influencing plant morphological, physiological, and biochemical processes [12,14,59]. Currently, some studies have indicated that the separate and/or the combined application of AMF with organic and/or inorganic amendments protects plants against the oxidative effects caused by environmental stresses [33,60,61,62,63]. Nevertheless, the recovery process after a prolonged period of DS is less studied, and the role played by AMF combined with VC and/or RP in this process is almost unknown. In this paper, we report the impact of combining native AMF with RP on the growth, physiology, metabolites, and ROS accumulation of carob under DS and after rehydration in soil amended with VC. The results of the present study reported that carob seedlings inoculated with AMF and/or amended with RP and/or VC showed a greater increase in growth under WW and DS conditions. The combined application of AMF with RP and VC results in greater plant fitness compared to a separate application of these amendments regardless of the applied water regime. Furthermore, stopping irrigation for 12 days had a more pronounced negative effect on the growth and biomass accumulation in uninoculated and unamended seedlings than in those inoculated and/or amended. Furthermore, the results showed that the application of native AMF in soil amendment with VC and RP can mitigate the negative effects of water stress on plant growth. The combination of an AMF consortium with an organic biofertilizer had higher growth than the control strawberry seedlings [64]. Furthermore, Akensous et al. [65] reported that the application of a tripartite combination (AMF, compost, and RP) improved date palm seedling growth under WW and DS conditions. In another study, the combination of an AMF consortium with VC showed an increase in the above-ground and root biomass of quinoa seedlings compared with the combination with compost, particularly under the DS condition [33]. The improved growth and biomass of carob seedlings treated with AMF and/or VC and/or RP is probably due to the ability of AMF to acquire nutrients available in the applied organic and inorganic amendments, and also to the water uptake efficiency of plants subjected to DS [66,67,68]. In addition, the availability and enhanced uptake of P and/or N by AMF has a positive impact on the photosynthetic machinery, on the biosynthesis of primary metabolites, on membrane structure and transport, and on cell division and elongation, leading to increased growth and accumulation of root and shoot biomass in plants [69,70]. The results related to AMF root colonization showed that the dual application of VC and RP favored infectious propagules in carob seedlings under the DS condition. Juntahum and Boonlue [41] reported that root colonization of sugarcane seedlings inoculated with *Funneliformis mosseae* was higher in the presence of RP under the DS condition. Moreover, a study reported by Paymaneh et al. [71] demonstrated that the highest mycorrhizal root colonization in pistachio seedlings was recorded in the combination of the AMF consortium containing *Glomus* and *Funneliformis* genera with VC. Several studies have reported that DS strongly alters the survival and germination of mycorrhizal structures and inhibits the propagation of AMF hyphae [72,73,74]. Overall, the effectiveness of (in)-organic amendments in improving root colonization is often measured in terms of improved seedling growth and biomass under long-term DS and during REC [12,75,76,77].

Under the DS condition, improved nutrient and water uptake is generally considered to be the essential benefit that AMF root colonization brings to their host plant [78]. In addition, VC and/or RP are amendments commonly used to increase the availability of certain nutrients and the water retention capacity [66,79]. The association of AMF with (in)-organic amendments has been suggested to enhance the nutrient uptake capacity of plants under the DS condition [67,80]. Furthermore, the results of the present study showed that the addition of VC and RP to AMF resulted in elevated P uptake in carob seedlings compared to the control plants under DS and WW conditions. This enhancement corroborates the results found by Caravaca et al. [81] and Zou et al. [82], which indicated that P uptake is more critical in plants inoculated under the DS condition than under the WW one. In addition, the improvement in leaf P levels was greater in mycorrhizal carob seedlings grown in soil amended with VC and/or RP under prolonged DS and REC. Juntahum and Boonlue [41] reported that the synergistic relationships between AMF and RP affect soil P availability in the rhizosphere surrounding sugarcane roots at all growth stages. Application of VC and RP can improve nutrient status, particularly P, in the soil [66,75]. Furthermore, improved AMF root colonization can contribute to increased nutrient uptake by the external mycelium, which enables nutrients to be transferred to plants [83,84]. The increase in P in the leaves of carob seedlings treated with AMF + VC + RP is probably due to the presence of readily available P in the soil. Therefore, efficient root colonization by mycorrhizal structures on seedlings has been linked to the availability and mobilization rate of P from the soil to plant seedlings and helps the latter to explore, accumulate, and transport phosphate [85].

The results of Ψ_Leaf_ showed that carob seedlings treated with the different fertilizers had a significantly higher water status under WW and DS conditions. The combination of AMF with VC and RP was considerably higher among the other treatments, particularly under conditions of prolonged DS. This improvement in Ψ_Leaf_ may be due to improved root colonization by AMF under WW and DS conditions. Wu et al. [72] reported that mycorrhizal plants had a better attitude to maintain higher water status under the DS condition than non-mycorrhized plants, due to the presence of extraradical mycorrhizal hyphal networks that played an essential role in the uptake and transport of water from the soil to the plant. In addition, it has been reported that AMF mycelial hyphae can replace aquaporin activity in the roots of AMF-inoculated plants under DS [31]. The application of VC as an organic amendment improves the Ψ_Leaf_ of carob seedlings under the DS condition. Several studies have reported that organic amendments enhance the water-holding capacity of dry soils [4,33,58,86]. Liu et al. [87] and Boutasknit et al. [12] reported that the enhanced AMF root colonization resulted in faster recovery of plants after rehydration compared with uncolonized plants. This rapid absorption of water after recovery results from a more extensive root system and AMF hyphae, which can increase the surface area of exploration beyond the root zone, thus increasing the volume of water absorbed by the plant [2,88,89]. Augé [90] and Zhu et al. [91] have indicated that higher Ψ_Leaf_ levels in mycorrhized and amended plants can be beneficial in shifting water to leaf evaporative surfaces and further stomata opening under DS conditions.

gs was significantly reduced in carob seedlings under the DS condition compared with those under the WW condition. However, the triple combination (AMF + VC + RP) contributed to more significant increases in gs compared with carob seedlings subjected to DS and after recovery. This improvement in gs in mycorrhized carob seedlings may be due to increased hydraulic conductance, fungi-root uptake surface, osmotic adjustment, and/or enhanced mycorrhizal colonization [11,91,92]. In addition, mycorrhizal colonization and osmotic adjustment (organic and inorganic) were higher in inoculated and amended seedlings, which contributed to improved plant water status, as demonstrated by the less negative Ψ_Leaf_ values. Arfan-Ul-haq et al. [93] reported that the application of RP in combination with manure increased gs in wheat. Consequently, the tripartite combination could positively influence the regulation of guard cell movement and opening, and the maintenance of gas exchange under DS, thus enabling faster recovery of the physiological state after rehydration.

The results showed a reduction in Fv/Fm in seedlings exposed to prolonged DS, particularly in uninoculated and unamended carob seedlings. However, carob seedlings inoculated with AMF and amended with VC and RP helped in maintaining greater PSII efficiency (Fv/Fm) under the DS condition and after recovery. In fact, the application of AMF in combination with VC and RP maintained and maximized the efficiency of excitation energy capture by chloroplasts and decreased damage to photosystem reaction centers under DS and recovery [94,95]. Some studies have shown that, under prolonged DS, mycorrhizal plants increased the maximum quantum yield of PSII [3,96,97]. Ye et al. [98] found that the Fv/Fm of grapevine plants inoculated with an AMF consortium was improved by 40% compared to the control plants after exposure to DS. In our study, the application of AMF consortium in combination with (in)-organic amendments resulted in a 30% improvement in Fv/Fm after 12 days of severe DS. In addition, soil enrichment with elements such as carbon, P, and N improved the metabolic and photosynthetic efficiency of plants under DS and after recovery. Abd El-Mageed et al. [99] reported that the application of a N-rich organic amendment led to an increase in Fv/Fm in eggplants exposed to DS. In addition, efficient P uptake played a crucial role in photosynthesis and carbohydrate synthesis. The combined application of AMF with VC and RP probably improved Fv/Fm quantum yield as it contributed to the efficient uptake of essential nutrients such as P and N under DS and REC conditions.

Prolonged DS has been shown to adversely affect the size of photosynthetic pigments in the reaction center [100,101]. Based on our results, the triple combination AMF + VC + RP was more effective in protecting chlorophyll pigments and leaf carotenoids in carob seedlings against degradation under DS. Wahid et al. [102] reported that the combination of AMF with an organic amendment and RP increased the content of photosynthetic pigments in green bean seedlings, compared with the control, which was explained by the effect of symbiotic N fixation and also by P assimilation by AMF [103]. Similarly, studies reported by Akensous et al. [65] showed that the application of AMF in combination with compost and RP improved the chlorophyll content of date palm plants under the DS condition [104,105]. Boutasknit et al. [106] reported that combining AMF with compost reduced oxidative damage and improved chlorophyll and carotenoid content under DS and after recovery. On the other hand, our results show that mycorrhized and amended seedlings accumulated more TSS and protein under DS conditions and after REC compared to the control seedlings. Yooyongwech et al. [3] reported that plants inoculated with AMF increased osmotic adjustment via the accumulation of high TSS and protein contents to maintain cell turgor and to cope with oxidative damage induced by DS. Abbaspour et al. [107] suggested that osmotic adjustment in mycorrhized plants was more active than in non-mycorrhized plants under WW and DS conditions. Ahanger et al. [108] recorded that the application of nano-VC resulted in a significant improvement (69.5%) in sugar content in tomato seedlings subjected to the DS condition compared to the control. Similarly, Benaffari et al. [34] reported that quinoa seedlings inoculated with AMF and amended with VC accumulated high levels of TSS under DS conditions. Similarly, protein content was higher in mycorrhizal seedlings grown in soil amended with VC and RP, which may explain the strengthening of the non-enzymatic antioxidant defense system under DS and after recovery conditions. The accumulation of organic osmolytes is linked to plant adaptation to water-stressed conditions, since the produced photosynthates and plant hormones can adjust and reduce osmotic potential at leaf level, which consequently leads to improved water uptake under the DS condition. In addition, increased concentrations of organic (TSS and proteins) and inorganic osmolytes participate in the protection of cellular integrity against oxidative damage and the accumulation of ROS in plant cells under environmental stress [109]. Exposure of plants to stress induces the production and accumulation of ROS at the cellular level, which causes the degradation and alteration of membrane lipids resulting in increased MDA levels [110,111]. The results of the present work indicate that during the DS condition, H_2_O_2_ and MDA concentrations are higher in the control seedlings compared with the inoculated and amended ones. The combination of AMF with VC and RP reduced H_2_O_2_ and MDA concentrations under DS and REC conditions. Indeed, the reduced levels of these molecules in inoculated and amended seedlings subjected to DS indicate their tolerance to oxidative stress [7,24,112,113]. This reduction in plants treated with AMF + VC + RP could be due to improved hydraulic conductivity and osmolyte levels, as well as higher antioxidant defense systems under DS [7,12,114,115]. In addition, inoculated and amended seedlings were associated with lower H_2_O_2_ and MDA accumulation, indicating lower oxidative damage in the cell membranes of these seedlings under stress and after recovery conditions. This hemostasis recorded in plants inoculated with AMF and (in)-organic amendments showed rapid recovery from the initial state with low levels of H_2_O_2_ and MDA after DS, indicating improved antioxidant enzyme activity and consequently improved tolerance of carob seedlings. Our results indicate that seedlings treated with AMF + VC + RP showed a significant increase in antioxidant enzyme activity (SOD, POX, and PPO) compared with the controls particularly under DS and after recovery. Similar results were reported by Boutasknit et al. [58], who showed a significant increase in antioxidant enzymes (POX and PPO) in carob seedlings inoculated with AMF and/or amended with compost compared to the control plants subjected to DS. It has been reported that increased antioxidant enzyme activities promote ROS elimination, leading to greater tolerance to water-limited conditions in plants [116]. In addition, plants with higher levels of SOD and POX activity were protected against DS-induced oxidative damage [117,118]. This protection is due to the role of the SOD enzyme in the conversion of ROS to H_2_O_2_ and the role of CAT and POX in the conversion of H_2_O_2_ to O_2_ and water [119].

## 5. Conclusions

In this study, we demonstrated that prolonged water stress causes considerable effects on growth and biomass accumulation, affecting the acquisition of water and mineral nutrients, the photosynthetic pigment content, and the accumulation of osmolytes as well as the stability of the cell membrane of carob seedlings. However, the application of AMF in combination with (in)-organic amendments made it possible to attenuate the damage induced by DS by promoting the water status, photosynthetic pigment contents, osmoregulation, and antioxidant defense system, thus preserving membranes and cellular components against oxidative damage. Indeed, the maintenance of physiological and biochemical functions in carob seedlings treated with AMF, VC, and RP can mitigate photosynthesis and ionic homeostasis disruption and protect membrane stability to resist a prolonged period of water stress. This will quickly recover the physiological state and the water and metabolic homeostasis of the seedlings after rehydration. These results highlight the importance of inoculating carob plants with a combination of AMF and (in)organic amendments, in particular, VC and RP, as an effective strategy for improving adaptation and tolerance to drought stress and facilitating robust recovery after rehydration, especially in the context of irregular and insufficient rainfall patterns prevalent in arid and semi-arid regions throughout the year. Consequently, this intelligent management of natural resources can be considered an important biotechnological method for ensuring the success of carob reforestation in forests subject to prolonged water stress.

## Figures and Tables

**Figure 1 metabolites-14-00202-f001:**
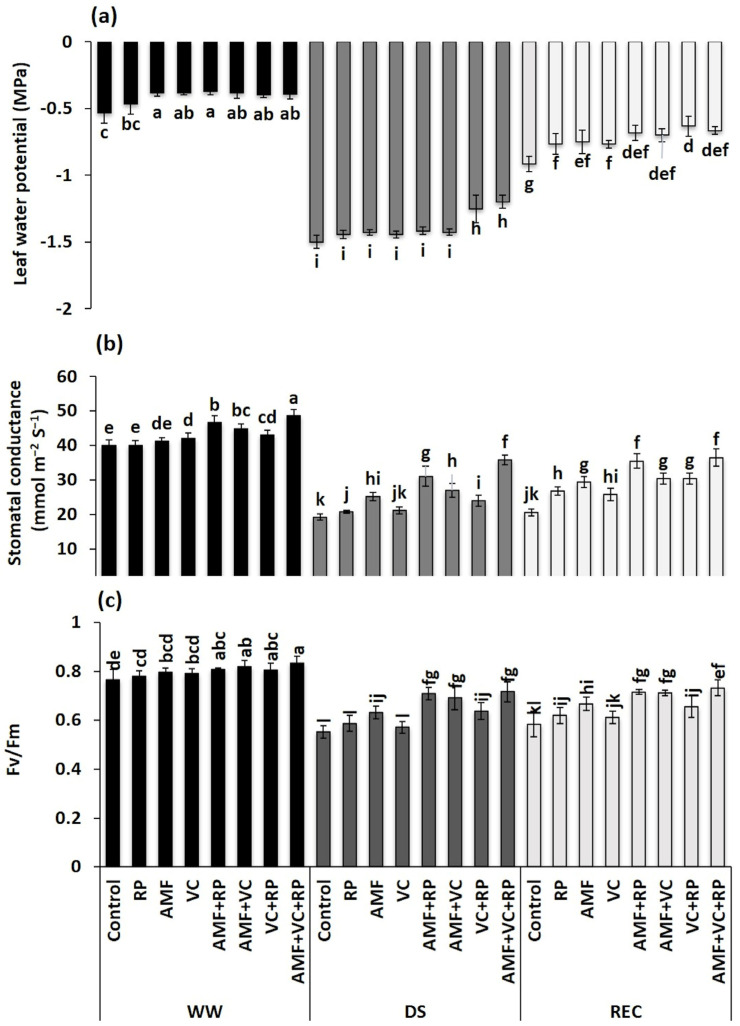
Effect of different combinations of AMF, RP, and VC on leaf water potential (**a**), stomatal conductance (**b**), and chlorophyll fluorescence (Fv/Fm) (**c**) of carob seedlings under different water regimes (WW: well-watered, DS: drought stress, and REC: recovery). AMF: arbuscular mycorrhizal fungi, VC: vermicompost, RP: rock phosphate. Different letters within each line denote significant differences at *p* < 0.05 based on Tukey’s test.

**Figure 2 metabolites-14-00202-f002:**
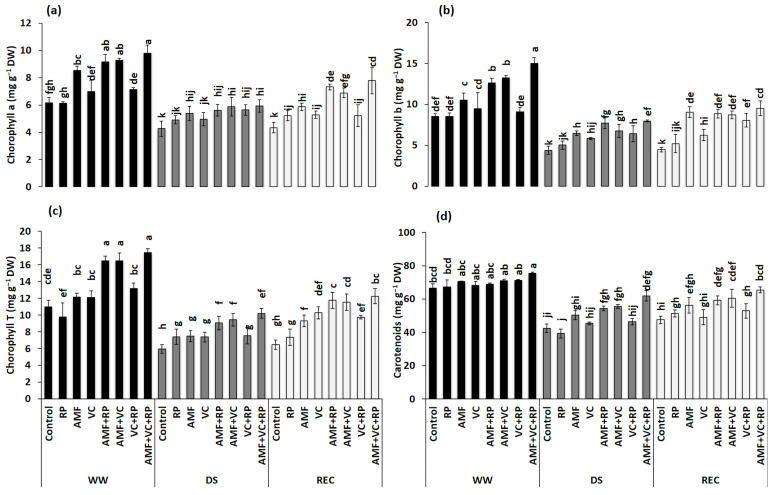
Effect of different combinations of AMF, RP, and VC on chlorophyll a (**a**), chlorophyll b (**b**), total chlorophyll (**c**), and carotenoid (**d**) content of carob seedlings under different water regimes (WW: well-watered, DS: drought stress, and REC: recovery). AMF: arbuscular mycorrhizal fungi, VC: vermicompost, RP: rock phosphate. Different letters within each line denote significant differences at *p* < 0.05 based on Tukey’s test.

**Figure 3 metabolites-14-00202-f003:**
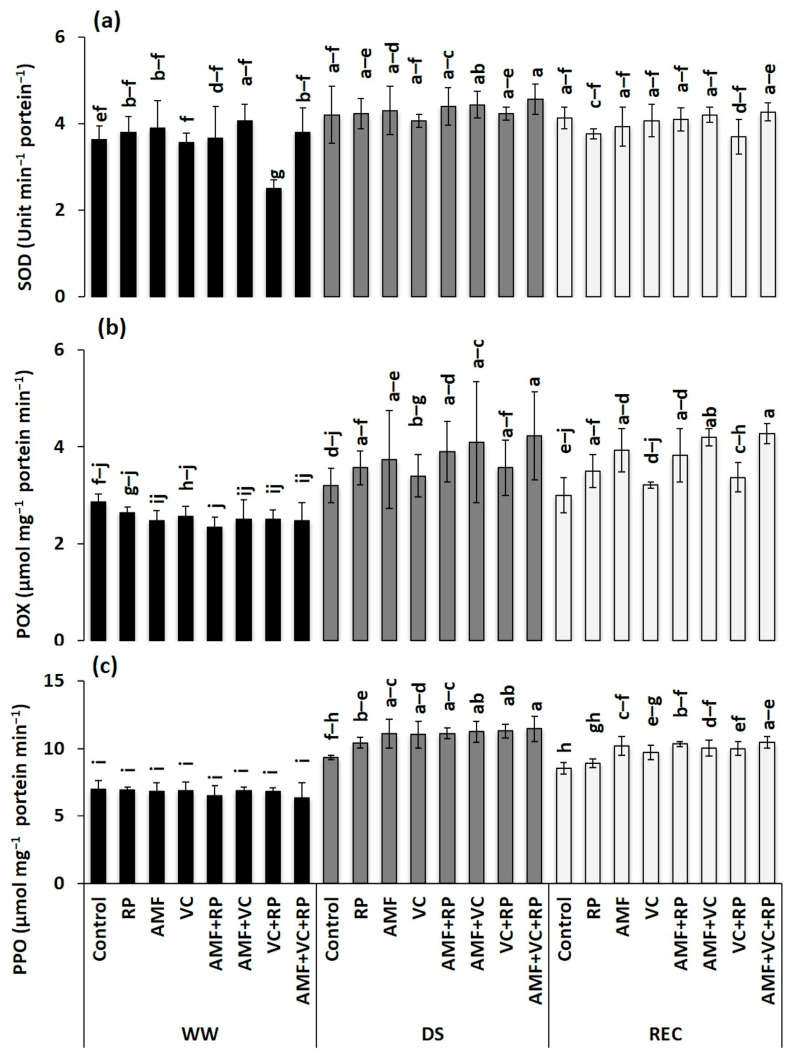
Effect of different combinations of AMF, RP, and VC on superoxide dismutase (SOD) (**a**), peroxidase (POX) (**b**), and polyphenol oxidase (PPO) (**c**) of carob seedlings under different water regimes (WW: well-watered, DS: drought stress, and REC: recovery). AMF: arbuscular mycorrhizal fungi, VC: vermicompost, RP: rock phosphate. Different letters within each line denote significant differences at *p* < 0.05 based on Tukey’s test.

**Table 1 metabolites-14-00202-t001:** Physicochemical properties of the vermicompost.

pH	EC (mS cm^−1^)	OM (%)	TN (%)	AP (mg Kg^−1^)	K (mg Kg^−1^)	Ca (mg Kg^−1^)	Fe (mg Kg^−1^)	Zn (mg Kg^−1^)	Cu (mg Kg^−1^)
7.70	0.14	43.80	0.62	290	4960	1900	820.10	44.90	27.40

AP: available phosphorus, Ca: calcium, Cu: copper, EC: electrical conductivity, Fe: iron, K: potassium, OM: organic matter, TN: total nitrogen, Zn: zinc.

**Table 2 metabolites-14-00202-t002:** Results of MANOVA for independent variables including drought, recovery, inoculation with AMF, and amendment with vermicompost and rock phosphate and the interaction among them.

Parameters	AMF (A)	VC (B)	RP (C)	DS (D)	REC (E)	A × D	A × E	B × D	B × E	C × D	C × E	A × B × D	A × B × E	A × C × D	A × C × E	B × C × D	B × C × E	A × B × C × D	A × B × C × E
F	***	**	***	***	***	***	***	ns	*	*	*	ns	*	*	*	ns	ns	ns	ns
I	***	ns	ns	***	***	***	***	ns	ns	ns	ns	ns	***	ns	ns	ns	ns	ns	ns
SH	***	***	***	ns	*	ns	ns	ns	ns	ns	ns	ns	***	ns	ns	ns	ns	ns	ns
RL	***	***	***	ns	**	ns	ns	ns	ns	ns	ns	ns	***	ns	ns	ns	ns	ns	ns
SDW	***	***	***	***	**	ns	ns	ns	ns	ns	ns	ns	***	ns	ns	ns	ns	ns	ns
RDW	***	***	***	**	ns	ns	ns	ns	ns	ns	ns	ns	***	ns	ns	ns	ns	ns	ns
gs	***	***	***	***	***	***	***	ns	ns	***	***	ns	**	ns	*	ns	ns	ns	ns
Fv/Fm	***	***	***	***	***	***	***	ns	ns	**	*	ns	ns	ns	ns	ns	ns	ns	ns
Ψ_Leaf_	***	***	***	***	***	ns	ns	*	ns	***	**	ns	ns	ns	ns	***	ns	ns	ns
Chl a	***	***	***	***	***	***	*	ns	ns	ns	ns	ns	ns	ns	ns	ns	ns	ns	ns
Chl b	***	***	***	***	***	***	*	ns	ns	ns	ns	***	***	ns	***	ns	ns	ns	ns
Chl T	***	***	***	***	***	***	**	**	ns	ns	ns	ns	*	*	ns	ns	ns	***	**
Car	***	***	**	***	***	***	***	ns	ns	ns	ns	ns	ns	*	ns	ns	ns	ns	ns
TSS	***	**	***	***	***	ns	ns	ns	ns	ns	ns	ns	ns	ns	ns	ns	ns	ns	ns
Proteins	***	***	***	***	***	**	*	ns	ns	ns	ns	ns	ns	ns	ns	ns	*	ns	ns
H_2_O_2_	***	*	*	***	**	ns	*	**	*	ns	ns	**	***	ns	ns	ns	*	ns	ns
MDA	***	ns	ns	***	***	ns	ns	ns	ns	ns	ns	ns	ns	ns	ns	ns	*	ns	ns
SOD	*	ns	ns	***	***	ns	ns	ns	ns	*	ns	ns	ns	ns	ns	ns	ns	ns	ns
PPO	***	***	ns	***	***	**	***	*	ns	ns	ns	ns	ns	ns	ns	ns	ns	ns	ns
POX	***	ns	ns	***	***	**	***	ns	ns	ns	ns	ns	ns	ns	ns	ns	ns	ns	ns
P	*	***	***	*	ns	ns	ns	ns	ns	ns	ns	ns	ns	ns	ns	ns	ns	ns	ns

AMF: arbuscular mycorrhizal fungi, VC: vermicompost, RP: rock phosphate, DS: drought stress, REC: recovery, F: mycorrhizal colonization frequency, I: mycorrhizal colonization intensity, SH: shoot height, RL: root length, SDW: shoot dry weight, RDW: root dry weight, Ψ_Leaf_: leaf water potential, gs: stomatal conductance, Fv/Fm: chlorophyll fluorescence, Chl: chlorophyll, Chl T: total chlorophyll, Car: carotenoid, TSS: total soluble sugar, H_2_O_2_: hydrogen peroxide, MDA: malondialdehyde, SOD: superoxide dismutase, POX: peroxidase, PPO: polyphenol oxidase, P: phosphorus. ns, not significant, * *p* < 0.05, ** *p* < 0.01, *** *p* < 0.001.

**Table 3 metabolites-14-00202-t003:** Effect of different combinations of AMF, RP, and VC on mycorrhizal root colonization, growth parameters, and phosphorus content of carob seedlings under different water regimes.

		F (%)	I (%)	SH (cm)	RL (cm)	SDW (g)	RDW (g)	P (mg g^−1^)
WW	Control	0.0 ± 0.0 f	0.0 ± 0.0 e	16.1 ± 0.9 j	33.3 ± 1.3 i	1.6 ± 0.4 l	0.9 ± 0.0 n	2.0 ± 0.4 f
	RP	0.0 ± 0.0 f	0.0 ± 0.0 e	23.0 ± 2.0 j	42.3 ± 1.9 gh	2.2 ± 0.3 k	1.2 ± 0.2 i–n	2.7 ± 0.4 b–e
	AMF	62.2 ± 5.1 bcd	51.3 ± 6.8 d	24.7 ± 1.8 gh	42.1 ± 1.4 h	2.5 ± 0.1 h–k	1.4 ± 0.3 h–k	2.4 ± 0.2 d–f
	VC	0.0 ± 0.0 f	0.0 ± 0.0 e	25.2 ±1.5 f–h	43.3 ± 2.5 f–h	2.3 ± 0.4 jk	1.1 ± 0.1 k–n	2.6 ± 0.4 c–f
	AMF + RP	61.0 ± 3.5 cd	62.3 ± 4.5 cd	27.1 ± 1.2 b–d	45.1 ± 1.0 d–f	3.1 ± 0.2 fg	1.7 ± 0.2 e–g	2.9 ± 0.5 a–d
	AMF + VC	53.8 ± 6.8 e	59.8 ± 4.4 ab	27.0 ± 0.5 b–d	47.2 ± 2.6 b–d	3.4 ± 0.3 d–f	1.9 ± 0.4 de	3.0 ± 0.5 a–d
	VC + RP	0.0 ± 0.0 f	0.0 ± 0.0 e	26.2 ± 0.5 d–f	45.9 ± 2.6 c–e	2.6 ± 0.5 hi	1.2 ± 0.4 i–m	3.1 ± 0.2 a–c
	AMF + VC + RP	53.7 ± 5.4 e	54.0 ± 7.6 cd	35.3 ± 1.3 a	52.8 ± 2.6 a	3.6 ± 0.2 b–e	2.4 ± 0.3 ab	3.2 ± 0.4 a–c
DS	Control	0.0± 0.0 f	0.0 ± 0.0 e	16.1 ± 0.7 j	34.1 ± 2.0 i	1.8 ± 0.3 l	1.0 ± 0.2 mn	2.1 ± 0.2 ef
	RP	0.0 ± 0.0 f	0.0 ± 0.0 e	23.8 ± 1.3 i	42.0 ± 0.5 gh	2.7 ± 0.2 hi	1.3 ± 0.3 i–m	3.0 ± 0.2 a–d
	AMF	64.5 ± 4.1 abc	58.0 ± 7.6 bc	24.7 ± 1.2 gh	42.3 ± 1.5 gh	2.8 ± 0.2 gh	1.4 ± 0.3 g–i	2.8 ± 0.2 a–d
	VC	0.0 ± 0.0 f	0.0 ± 0.0 e	25.4 ± 1.4 e–h	43.3 ± 2.4 f–h	2.6 ± 0.3 h–j	1.3 ± 0.2 i–l	3.1 ± 0.3 a–c
	AMF + RP	66.3 ± 5.8 ab	62.3 ± 4.5 ab	27.1 ± 0.8 b–d	45.9 ± 1.2 b–e	3.7 ± 0.3 b–d	1.6 ± 0.2 f–h	3.2 ± 0.2 a–c
	AMF + VC	54.5 ± 8.0 e	61.5 ± 1.6 ab	27.7 ± 0.6 bc	47.9 ± 2.3 bc	3.9 ± 0.4 a–c	2.1 ± 0.3 cd	3.2 ± 0.4 a–c
	VC + RP	0.0 ± 0.0 f	0.0 ± 0.0 e	26.1 ± 0.9 d–f	45.8 ± 1.3 c–e	3.2 ± 0.3 f	1.4 ± 0.4 g–i	3.1 ± 0.3 a–d
	AMF + VC + RP	66.5 ± 6.9 a	61.8 ± 4.4 ab	35.3 ± 0.8 a	53.1 ± 1.4 a	4.1 ± 0.2 a	2.7 ± 0.2 a	3.3 ± 0.4 ab
REC	Control	0.0 ± 0.0 f	0.0 ± 0.0 e	16.5 ± 1.1 j	34.8 ± 0.7 i	1.7 ± 0.3 l	0.9 ± 0.2 n	2.0 ± 0.4 ef
	RP	0.0 ± 0.0 f	0.0 ± 0.0 e	25.0 ± 1.0 hi	43.5 ± 1.8 i	2.3 ± 0.2 i–k	1.1 ± 0.0 l–n	2.9 ± 0.5 a–d
	AMF	61.8 ± 4.4 cd	60.7 ± 4.1 ab	25.3 ± 0.8 e–h	43.1 ± 2.6 f–h	2.6 ± 0.4 hi	1.4 ± 0.3 g–j	2.9 ± 0.5 a–d
	VC	0.0 ± 0.0 f	0.0 ± 0.0 e	25.6 ± 1.4 e–g	44.5 ± 1.3 e–g	2.4 ± 0.3 i–k	1.2 ± 0.1 i–n	3.0 ± 0.3 a–d
	AMF + RP	67.2 ± 4.5 a	63.7 ± 3.1 a	27.6 ± 0.5 bc	46.5 ± 1.1 b–e	3.3 ± 0.4 ef	1.6 ± 0.3 f–h	3.4 ± 0.3 a
	AMF + VC	58.8 ± 3.2 d	62.0 ± 2.4 ab	27.8 ± 0.5 b	48.1 ± 2.9 b	3.6 ± 0.5 c–e	1.9 ± 0.3 d–f	3.1 ± 0.6 a–c
	VC + RP	0.0 ± 0.0 f	0.0 ± 0.0 e	26.5 ± 0.8 c–e	46.8 ± 2.6 b–d	2.8 ± 0.3 gh	1.3 ± 0.2 i–l	3.0 ± 0.2 a–d
	AMF + VC + RP	66.7 ± 3.5 a	62.2 ± 2.8 a	35.8 ± 1.0 a	53.8 ± 1.6 a	3.9 ± 0.3 ab	2.3 ± 0.2 bc	3.3 ± 0.6 a–c

AMF: arbuscular mycorrhizal fungi, VC: vermicompost, RP: rock phosphate, AMF + RP: AMF consortium and rock phosphate combination, AMF + VC: AMF consortium and vermicompost combination, VC + RP: vermicompost and rock phosphate combination, AMF + VC + RP: AMF consortium, rock phosphate and vermicompost combination. SH: shoot height, RL: root length, SDW: shoot dry weight, RDW: root dry weight, P: phosphorus, WW: well-watered, DS: drought stress, REC: recovery. Means (±standard error) within the same parameter, followed by different letters, are significantly different among treatments at *p* ≤ 0.05.

**Table 4 metabolites-14-00202-t004:** Effect of different combinations of AMF, RP, and VC on total soluble sugar, proteins, hydrogen peroxide, and malondialdehyde content of carob seedlings under different water regimes.

		TSS (mg g^−1^)	Proteins (mg g^−1^)	H_2_O_2_ (µmol g^−1^ DW)	MDA (µmol g^−1^ DW)
WW	Control	12.6 ± 0.8 k	11.5 ± 0.9 cd	14.3 ± 0.6 c–f	12.3 ± 0.8 c–h
	RP	14.7 ± 0.9 h–j	11.5 ± 0.7 cd	14.3 ± 1.3 c–f	11.3 ± 0.6 g–j
	AMF	14.7 ± 0.7 h–j	11.7 ± 0.2 bc	13.4 ± 1.2 fg	11.8 ± 0.5 f–j
	VC	13.4 ± 1.9 jk	11.8 ± 0.7 bc	16.3 ± 2.1 b	11.0 ± 0.6 ij
	AMF + RP	14.3 ± 0.4 i–k	12.0 ± 0.4 bc	12.9 ± 0.5 fg	10.6 ± 1.6 c–h
	AMF + VC	16.1 ± 0.8 gh	12.1 ± 0.8 bc	14.3 ± 1.0 c–f	11.3 ± 1.0 h–j
	VC + RP	14.3 ± 1.0 ij	12.6 ± 1.5 ab	15.1 ± 1.2 b–e	12.4 ± 0.5 c–h
	AMF + VC + RP	16.3 ± 0.7 f–h	13.4 ± 1.5 a	12.2 ± 0.4 g	11.5 ± 1.2 g–j
DS	Control	17.8 ± 0.8 c–f	7.8 ± 0.5 k	19.7 ± 1.3 a	14.4 ± 0.5 a
	RP	19.4 ± 1.5	8.8 ± 0.8 hi	19.6 ± 1.0 a	14.0 ± 0.8 ab
	AMF	19.3 ± 1.4 a–c	9.9 ± 0.1 jk	16.5 ± 0.8 b	13.5 ± 0.9 a–d
	VC	19.1 ± 1.3 a–d	8.5 ± 0.4 d–h	19.4 ± 0.7 a	14.1 ± 0.3 ab
	AMF + RP	19.6 ± 1.0 ab	10.4 ± 0.2 f–h	15.5 ± 1.1 b–d	13.2 ± 0.8 a–e
	AMF + VC	19.0 ± 0.5 a–d	10.1 ± 0.4 f–h	15.7 ± 0.6 bc	13.4 ± 1.2 a–d
	VC + RP	19.9 ± 0.8 ab	9.4 ± 0.7 h–j	16.1 ± 0.3 b	14.1 ± 0.8 ab
	AMF + VC + RP	20.2 ± 0.7 a	11.0 ± 0.1 c–g	15.3 ± 1.1 b–e	13.0 ± 0.9 b–f
REC	Control	15.2 ± 0.4 hi	8.6 ± 0.5 jk	18.7 ± 1.2 a	13.3 ± 0.8 a–d
	RP	17.6 ± 0.5 d–g	10.2 ± 0.1 e–h	18.3 ± 0.4 a	13.9 ± 0.6 ab
	AMF	18.7 ± 0.4 a–e	10.0 ± 0.1 gh	13.7 ± 0.6 e–g	12.7 ± 0.6 b–g
	VC	17.2 ± 1.1 e–f	9.7 ± 0.2 hi	14.3 ± 1.1 c–f	13.6 ± 0.9 a–c
	AMF + RP	19.6 ± 0.9 ab	11.2 ± 0.5 c–f	12.1 ± 1.2 g	11.8 ± 0.5 e–i
	AMF + VC	19.0 ± 1.0 a–d	11.4 ± 0.4 cd	14.1 ± 0.6 d–f	12.1 ± 1.3 d–i
	VC + RP	18.5 ± 2.1 b–e	10.3 ± 0.5 e–h	16.4 ± 0.8 b	12.7 ± b–h
	AMF + VC + RP	20.1 ± 0.3 ab	11.3 ± 0.4 c–e	14.1 ± 0.5 c–f	11.5 ± 1.0 g–j

AMF: arbuscular mycorrhizal fungi, VC: vermicompost, RP: rock phosphate, TSS: total soluble sugars, MDA: malondialdehyde, WW: well-watered, DS: drought stress, REC: recover. Means (±standard error) within the same parameter, followed by different letters, are significantly different among treatments at *p* ≤ 0.05.

## Data Availability

The raw data supporting the conclusions of this article will be made available by the authors on request.

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
