# Peer review of "Effect of Arbuscular Mycorrhizal Fungi and Rock Phosphate on Growth, Physiology, and Biochemistry of Carob under Water Stress and after Rehydration in Vermicompost-Amended Soil"

_metabolites, 2024, doi:10.3390/metabo14040202_

Round 1
Reviewer 1 Report
Comments and Suggestions for Authors
Abderrahim Boutasknit et al
Effect of arbuscular mycorrhizal fungi and rock phosphate on growth, physiology and biochemistry of carob under water stress and after rehydration in vermicompost-amended soil
The project of Boutasknit et al., is aimed to improve the prospects for the reforestation of desert regions by carob tree.
To this end, Boutasknit et al., have tested different conditions to select the technological settings under which carob tree seedlings could survive long periods of drought without loss of productivity.
Article of Boutasknit et al. is an extensive study performed by the methods of classical plant physiology. The authors investigated the response of carob trees to drought, and tested methods that may improve the plant's survival efficiency under stress. They have used addition of vermicompost and rock phosphate, as well as different regimes of watering. As a most Authors analyzed the effect of growing plants in symbiosis with arbuscular micorriza, that Authors have used in form of a consortium of a mixture of 26 species of Glomales. This is interesting approach that enhances the possibility to model the stress response under near-natural conditions, since under natural conditions the presence of multiple microbial species is more likely than a single one.
Boutasknit et al. compared multiple variations of experimental conditions and have evaluated several parameters, including growth of the plants, the level of Root Mycorrhizal Colonization, leaf mineral phosphorus content, leaf water potential, stomatal conductance, chlorophyll fluorescence (Fv/Fm), total soluble sugars, total soluble protein and antioxidant enzyme activities, the activity of antioxidant enzymes superoxide dismutase, peroxidase, polyphenoloxidase.
As a result, Boutasknit et al. showed that such factors of vermicompost and rock phosphate as addition to the soil, and the symbiotic conditions were sufficient for survival and growth under prolonged ( 12 days) drought stress. The growth of plants was superior in case when all these conditions were combined, regardless of the applied water regime. А positive effect was also observed on such plant characteristics as leaf water potential and stomatal conductance, which could be expected since the positive effect of arbuscular mycorrhiza is largely due to the improved water balance of the plant.
As a result, Boutasknit et al. identified technological conditions under which, even in the absence of irrigation, plants could tolerate stress and grow. I believe that this set of measures can be a useful methodological tool for local farmers and businesses that grow carob trees.
I also suggest to make some corrections that are marked as “sticky notes” in PDF file I attach.
The authors performed extensive research that can be used in local agriculture. The advantages of the work include the use of classical, well-developed and cheap analysis methods, which will help repeat these analyzes if it is necessary to adapt other varieties or types of arbuscular mycorrhiza.
The paper can be accepted after minor revision.

I don't have any advice for improving English in the article.
Author Response
Please find attached our response to the comments of Reviewer 1

Reviewer 2 Report
Comments and Suggestions for Authors
Review report
The manuscript titled: Effect of arbuscular mycorrhizal fungi and rock phosphate on growth, physiology and biochemistry of carob under water stress and after rehydration in vermicompost-amended soil, by Boutasknit et al., describes The effects of arbuscular-mycorrhizal-fungi (AMF), vermicompost (VC) and rock phosphate (RP) on carob growth under drought stress (DS) for 12 days and for rapid recovery after rehydration (REC). Although a lot of work has been achieved, a serious revision is needed and many points need to be resolved before further processing.
L27: write drought stress before (DS)
L27: change (theses) to (these)
L30: what do you mean with water regimes favorable (WW)? Do you think that WW is the appropriate abbreviation? WW is the abbreviation of well-watered.
L50: change (droughts) to (drought)
L51: change (the functioning) to (the function)
L60-64: Long sentence, please divide into two sentences.
L71: livestock feed systems agroforesters? Do you think, it’s a correct expression?
L84: change (and) with (based on)
L95: change (improve) to (improves)
L108 to L116: the same meaning is repeated, please re-write to ignore the same meaning.
L123: which variety of Carob did you use? Mention the name of the variety.
L129: 568 mg g-1 available K, do you think you added 568 mg per gram? Or per kilogram?
L141-142: by passing 60 g of mycorrhizal inoculum through 20 ml of distilled water? I think its petter to change (through) to (dissolved in).
L147: under the table 1, add the full name (Zinc) of Zn abbreviation
L151: change (The RP used was) to (The used RP was)
L164-165: Authors applied water shortage for 12 days one time only, why you did not apply water shortage for several times (you could apply water stress for 2-4 months), Carob trees are very drought tolerant. do you think one time of applying drought stress is enough to measure the impact of yours treatments? Clarify?
L180: what is chloridric acid? Do you mean Hydrochloric acid?!
L179-180: They were then acidified with …………. They were then acidified with 5% chloridric acid. Re-write to ignore repetition and clarify the meaning.
L183: Add (I) after (intensity)
L237: change (The supernatant obtained) to (The obtained supernatant)
L249: change (The solution used consisted) to (The used solution consisted)
L285: change (showed an improvement of 9% and 8% and 5% and 2% respectively) to (showed an improvement reached 9%, 8%, 5% and 2%, respectively,)
L298-300: the percentage increases in shoot dry weight (SDW) and root dry weight (RDW) were greater …….? Change to (the increment in shoot dry weight (SDW) and root dry weight (RDW) were greater …….).
L300: delete (and) after AMF.
L301: 125% vs. 122% and 176% vs. 169% respectively? Please clarify how did you calculate these percentages?
L328-329: carob plants inoculated with AMF and/or amended with VC and/or RP? Authors could write a better sentence to clear the meaning as follows: carob plants inoculated only with AMF or combined with VC and/or RP.
L332-333: Regardless of the treatment applied, carob seedlings significantly increased phosphorus levels? How can carob seedlings increase the P levels? Do you mean that treatments with AMF, RP and VC and their combinations that increased the P levels? Or carob plants themselves dissolve RP and increase the available P in the soil. Please clarify?
L336-337 and L339: Again, how did authors calculated the percentages of accumulation from phosphorus levels under DS and REC?
L347: Under DS water stress? Its wrong writing, should be as follows: Under drought stress (DS).
L365-368: The extent of gs decline was greater in control carob seedlings by 52% than in those inoculated with AMF and amended with RP (AMF+RP) and/or with VC (AMF+VC+RP) by 22% and 10% respectively, compared to the controls under WW conditions. I can not understand, how was the ratio 52% and then by 22% and 10%? Under which conditions are these percentages? How did the authors calculated these ratios? Clarify.
L396: grown under WW favorable watering conditions! Wrong style, re-write.
L407-409: This increase was 69%, 59% and 80% for Chl a, 99%, 95% and 114% for Chl b, 82%, 79% and 89% for Chl T, and 24%, 27% and 37% for carotenoids in AMF+RP, AMF+VC and AMF+VC+RP respectively. Again, authors here compared between treatments against control of WW or control of REC? how they calculated the increased percentages here? clarify.
LL429: water regime levels? Authors did not apply water regime levels, they applied one water stress level (for 12 days) for one time only? Again, why authors did not apply water stress for a period of time (for example for 2-4 months) with regular shortages in irrigation (water stress)? Because carob considers a drought tolerance plant.
L433-439: authors compared the TSS of AMF+RP, AMF+VC and AMF+VC+RP under DS with the control of DS whereas they compared the Proteins of the same treatments with the control of WW, why authors did not follow the same trend in their comparisons, its better to compare the results you obtained with the control of the same treatment, for example, you compare the TSS of treatments under DS with the control of DS, and so on.
H2O2 should be subscript in over all the manuscript
L483: Are you sure that AMF+VC+RP increased the SOD up to 260%? Revise.
L594-595: all scientific names must be in italic form, please follow in all over the manuscript.
The discussion section is too long and some information are repeated, authors should reduce it to the half.
L853-858: a long sentence that need to be divided to maintain the meaning.
The conclusion section also is long and need to be reduced.
Extensive English revision is needed and revision by native speaker is recommended.

Extensive English revision is required and revision by native speaker is recommended.
Author Response
Please find attached our response to the comments of Reviewer 2

Round 2
Reviewer 2 Report
Comments and Suggestions for Authors
Authors addressed all raised points and the manuscript after revision has been improved and can be accepted for publication after minor english check.
Comments on the Quality of English LanguageMinor revision in english language can be checked